# Multi-Sensor 3D Survey: Aerial and Terrestrial Data Fusion and 3D Modeling Applied to a Complex Historic Architecture at Risk

Marco Roggero [1,*] and Filippo Diara [2]

1 Department of Architecture and Design (DAD), Polytechnic of Turin, Viale P.A. Mattioli 39, 10125 Turin, Italy
2 Historical Studies Department, University of Turin, Via Sant'Ottavio 20, 10124 Turin, Italy; filippo.diara@unito.it
* Correspondence: marco.roggero@polito.it

**Abstract:** This work is inscribed into a more comprehensive project related to the architectural requalification and restoration of Frinco Castle, one of the most significant fortified medieval structures in the Monferrato area (province of Asti, Italy), that experienced a structural collapse. In particular, this manuscript focuses on data fusion of multi-sensor acquisitions of metric surveys for 3D documenting this structural-risky building. The structural collapse made the entire south front fragile. The metric survey was performed by using terrestrial and aerial sensors to reach every area of the building. Topographically oriented Terrestrial Laser Scans (TLS) data were collected for the exterior and interior of the building, along with the DJI Zenmuse L1 Airborne Laser Scans (ALS) and Zenmuse P1 Photogrammetric Point Cloud (APC). First, the internal alignment in the TLS data set was verified, followed by the intra-technique alignments, choosing TLS as the reference data set. The point clouds from each sensor were analyzed by computing voxel-based point density and roughness, then segmented, aligned, and fused. 3D acquisitions and segmentation processes were fundamental for having a complete and structured dataset of almost every outdoor and indoor area of the castle. The collected metrics data was the starting point for the modeling phase to prepare 2D and 3D outputs fundamental for the restoration process.

**Keywords:** 3D survey; data fusion; LiDAR; photogrammetry; TLS; UAS





## 1. Introduction

In complex and risky architectural contexts, multi-sensor data fusion overcomes the limitations of a single sensor and the time constraints for an investigation. It is a must-have solution for outdoor and indoor surveying, as well as for surveying roofs and surroundings. This study discusses the topic of the fusion of point clouds from different sensors and methodologies [1–3], optimally combining LiDAR scans both from terrestrial (4.5 billion points) and Unmanned Aerial Systems (UAS, 200 million points), as well as photogrammetric point clouds (350 million points), to obtain an improved and complete 3D model of a complex historic building at risk.

The analyzed building is Frinco Castle, as shown in Figure 1 (province of Asti, Italy). Despite its imposing medieval defensive structure, it suffered different structural weaknesses and collapsed—combining several buildings throughout the ages contributed to its architectural complexity.

Surveying Frinco Castle required different surveying techniques due to the morphological complexity of the building. The integrated GNSS/total station control network was set up around the castle, and many compound traverses were used to connect the inside survey to the external reference. A complete 3D scan of the castle was acquired by integrating the outside/inside terrestrial laser scanner (TLS) survey with UAV (Unmanned Aerial Vehicle)/UAS (Unmanned Aircraft System) flights.

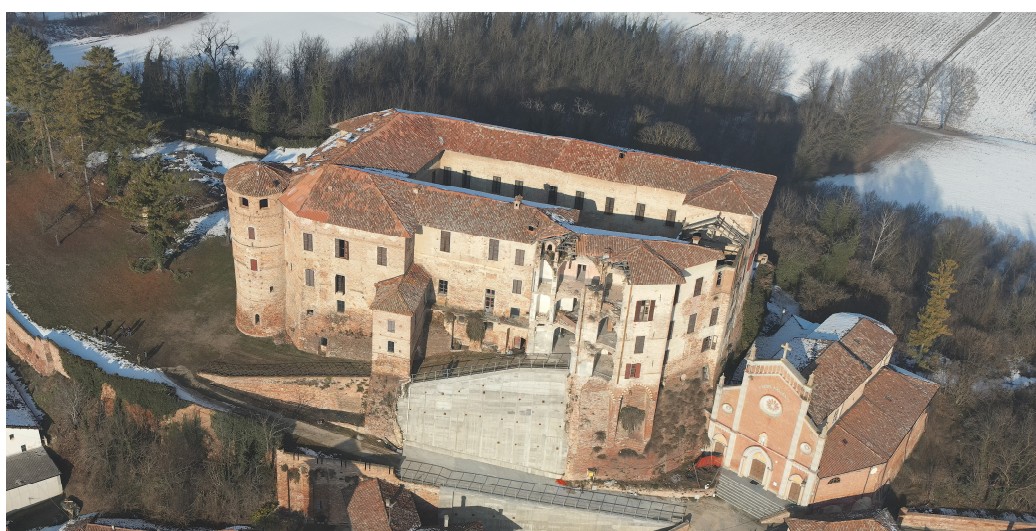

**Figure 1.** Frinco Castle from the UAS P1 sensor. Overview of the southwest side and the collapsed area.

The collected data generated from different sensors and methodologies represent the castle's global and punctual metric documentation. However, digital 2D and 3D outputs for the restoration process required a complete and homogeneous overview of point clouds. For this reason, the main goal of this work focuses on metric data refinement, alignment, fusion (from terrestrial and aerial sensors) and metric evaluation of these processes.

The segmentation and classification steps were also carried out in the indoor scans, where different object classes (floor, vault, wall, stairs, and wooden structures) can be selected and performed manually, automatically, or in an algorithm-assisted manual mode. The vaulted rooms related to the collapsed area, also surveyed by UAS, experienced mixed segmentation and alignment.

Finally, the fused metric data constituted the backbone for a parametric modeling design: in fact, initial and partial parametric modeling was performed for wooden beams on the rooftop (especially for the vaulted rooms of the collapsed area); then, NURBS modeling was conducted for having a lightweight model on which mapping stratigraphic features of the masonries and generating detailed photographic renders.

*The Case Study: The Medieval Castle of Frinco*

The surveyed context is related to the defensive medieval structure in Frinco municipality (AT-Italy). The castle is a fragile building resulting from architectural modifications since the 13th century [4–7]. In fact, architectural extensions and renovations (indoor and outdoor) have created a complex maze that is difficult to decode. The recent stratigraphic analysis [8] confirmed the initial nucleus of tower houses that experienced merging operations during the time. Architectural elements such as bichrome pointed arches, single lancet windows, and the Ghibelline decorations (dovetail battlements) represent important chronological markers related to a period between the 13th and the 15th centuries [8]. Though the medieval period is well-attested in the overall building stratigraphy, the most detectable phase is related to a chronological range between the latter half of the 18th century and the end of the 19th century [8]. The stratigraphic analysis was fundamental for understanding the evolution of the construction site as well as for having a detailed overview of fragile masonry units regarding the outlook for the reconstruction. In fact, a collapse in 2011 endangered the security of the castle and the houses beneath it. A large chunk of the manor collapsed into the populated area in 2014, flooding the churchyard and the public road and encircling the residences. The municipality bought the castle in 2019 and started consolidation and restoration projects in 2020. For the rehabilitation project,

the municipality of Frinco commissioned the castle's topographical and architectural 3D surveys in 2021.

## 2. Metric Survey and Quality Check

### 2.1. Surveying Equipment

The control network traverse surveying and the ground 3D scans were performed by the Trimble SX10 scanning station [9], with the advantage of handling a unique device on the field instead of operating a laser scanner and a total station. This instrument combines the topographic survey with LiDAR acquisitions, allowing a direct topographic point cloud orientation and avoiding scan registration at the office. It reaches an accuracy of 1 mm + 1.5 ppm with prism and 2 mm + 1.5 ppm di DR mode.

The aerial survey was operated by DJI Matrice 300 RTK equipped with DJI Zenmuse L1 LiDAR and P1 photogrammetric sensors [10]. An exhaustive comparison of terrestrial and aerial point clouds related to Frinco Castle 3D survey can be found in previous work [11]. Also, the reliability and accuracy of the UAS systems [12–14] and scanning stations [15] have been demonstrated by similar analysis.

### 2.2. Working Planning

Spanning 70 by 40 m and standing 30 m tall from its base to the rooftop, Frinco Castle is a grand structure comprising four primary levels and several intermediate floors. Given the building's intricate design and substantial size, the metric survey was carried out using the Trimble SX10 scanning station. This integrated total station allowed us to benefit from the direct alignment of the scanning stations via traverse adjustment, thereby enabling a seamless connection between the indoor and outdoor point clouds.

A closed traverse was established and examined to serve as a control network composed of eight GNSS points, which were then linked via a high-precision topographic survey (Figure 2). The Control Points (CPs), positioned in redundant numbers around the building area and used as reference points for all subsequent survey work, are stainless steel blocks stamped with the point's name.

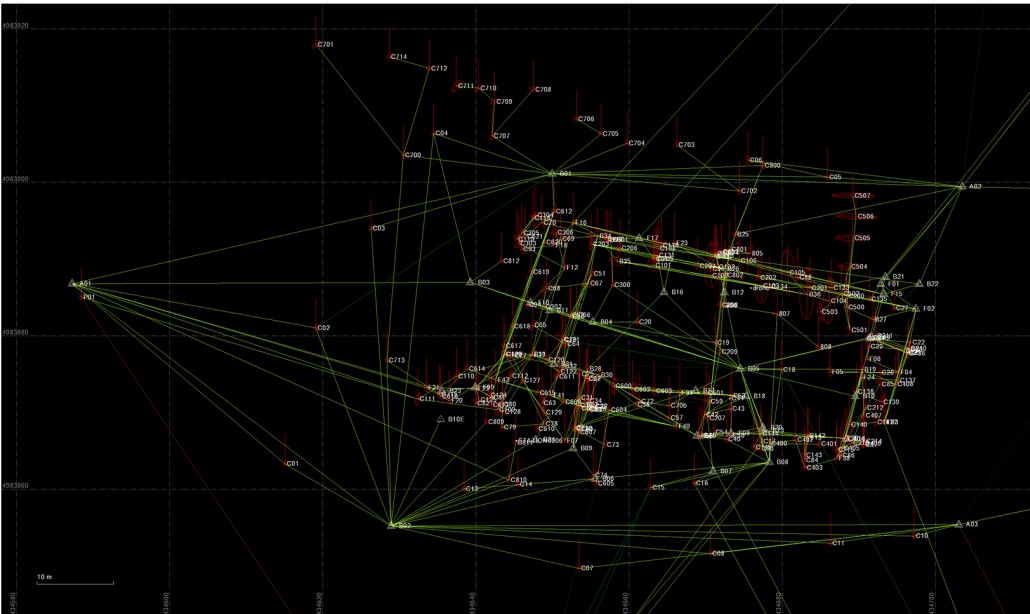

**Figure 2.** The Frinco Castle control network (Green) and CP error ellipses (Red).

Given the average dimensions of the castle's rooms, the expected scan density in standard mode indoors is one point every 2–6 mm, which is sufficient for the architectural survey's deliverables. The full-dome scan in coarse mode, coupled with the panorama

capture, takes approximately 15 min per scan station. Each scan station necessitates about 20–30 min, factoring in the landmarks set up by stainless steel studs, equipment relocation along a traverse, and station setup. Connecting an indoor traverse to the outdoor control network extends the survey duration due to the lengthy path required to transport the equipment from inside to outside and vice versa. Accessing some rooms, especially in the basement and attic, was occasionally challenging. The need to acquire more than 300 scans led to a 32-day survey campaign, meticulously planned using a Gantt Chart. During this campaign, 25 days were allocated explicitly for capturing scans, with an average of 12 scan stations per day. An additional seven days were required for other tasks: the aerial photogrammetric and LiDAR survey, the manual verification and photographic documentation of openings, doors, windows, and floors, and the photographic survey of the facades to produce high-resolution orthophoto deliverables.

### 2.3. Control Network Adjustment

The user can control SX10 with the Trimble Access™ (TA) field software (version 2023.00) operating on the Trimble Tablet Rugged PC. The tablet can communicate with the SX10 via radio, Bluetooth, or Wi-Fi. During scan acquisition, WIFI communication is mandatory due to the large amount of data to be transmitted, which requires a high bit rate. Optical, leveling, and GNSS data can be adjusted and processed in the office by Trimble Business CenterTM (TBC) software (version 2023.10). All Trimble software is available on the official website [9].

Throughout a 32-day campaign, the survey work was executed, sequentially adjusting the control network and various indoor traverses according to the following general workflow:

1. Data collection and surveying using Trimble Access™ (TA)
2. Exportation of *.job file and subsequent importation into Trimble Business Center™ (TBC)
3. Establishment of control points and execution of network adjustment
4. Optional exportation of scans to Trimble Realworks™ (version 12.0)
5. Repetition of steps 1–4 until the completion of the survey.

To check the adjustment quality, the final adjusted coordinates were extracted and RMS from TBC adjustment reports. Table 1 notices RMSs for each surveyed area: they are minimal outdoors, while much higher values are reported indoors due to often tricky operating conditions, such as confined spaces or nonstable floors, but above all due to the impossibility of closing some traverse.

**Table 1.** Control points RMSs of the complete control network.

| | | CP Number | Mean | | Max | |
|---|---|---|---|---|---|---|
| | | | $\sigma_{hor}$ | $\sigma_{vert}$ | $\sigma_{hor}$ | $\sigma_{vert}$ |
| Outdoor | | 15 | 0.005 | 0.002 | 0.009 | 0.003 |
| Indoor | Basement | 110 | 0.011 | 0.003 | 0.095 | 0.008 |
| | Level 1 | 52 | 0.008 | 0.003 | 0.022 | 0.008 |
| | Level 2 | 45 | 0.007 | 0.003 | 0.022 | 0.009 |
| | Garret | 45 | 0.015 | 0.002 | 0.044 | 0.004 |

Two sections of the structure, an underground icebox (12 CP) and a portion of the garret (13 CP) were surveyed by georeferencing the scans via open traverses. Upon excluding these two open traverses from the statistical analysis, the horizontal mean RMS values were found to be less than 10 mm, and the vertical mean RMS values were less than 3 mm. This data can be noticed in Table 2.

**Table 2.** Control points RMSs of the close traverse control network.

| | | CP Number | Mean | | Max | |
|---|---|---|---|---|---|---|
| | | | $\sigma_{hor}$ | $\sigma_{vert}$ | $\sigma_{hor}$ | $\sigma_{vert}$ |
| Outdoor | | 15 | 0.005 | 0.002 | 0.009 | 0.003 |
| Indoor | Basement | 98 | 0.008 | 0.003 | 0.022 | 0.008 |
| | Level 1 | 52 | 0.008 | 0.003 | 0.022 | 0.008 |
| | Level 2 | 45 | 0.007 | 0.003 | 0.022 | 0.009 |
| | Garret | 32 | 0.010 | 0.002 | 0.030 | 0.004 |

## 3. Point Cloud Alignment Methods Outdoor/Indoor

Cloud alignment can be achieved by setting standard targets in the scanned scene, which are typically spheres or reflective adhesive targets used to merge point clouds from adjacent stations. However, when standard targets are impossible or time-consuming, point cloud registration necessitates a cloud-to-cloud alignment that often employs a coarse-to-fine registration strategy. The initial registration parameters for the rigid body transformation of two-point clouds are estimated using a feature-based method. These features can encompass point features, building corners, flat walls or floors, pipes, and other similar site features. Surface feature-based methods are less susceptible to noise, but they demand substantial overlapping areas, with at least three pairs of surfaces needing to be present in the clouds to be registered.

In the fine registration phase, the primary goal is to maximize the overlap of two-point clouds, which is primarily achieved using the Iterative Closest Point method (ICP), RANdom SAmple Consensus method (RANSAC), Normal Distribution Transform method (NDT), or methods that utilize auxiliary data, such as GNSS scan position or standard targets. The most widespread process is the iterative approximation, which primarily refers to the ICP algorithm and a series of enhanced algorithms. However, this method presumes a reasonable estimation of the initial scan location and necessitates a high overlap between the two point clouds.

In cloud-to-cloud alignment, even using standard targets, one cloud is set as a reference, and one is moved to perform a pairwise registration. This causes the movable scan stations to align with the reference scan station, resulting in a station group that can be further refined if necessary. The aligned scan station moves to the reference scan group, and other scans are then aligned and sequentially moved to the reference group. Finally, the overall registration refinement can be performed. The method can be time-consuming and lead to misalignments, due to alignment error propagation, especially in indoor environments and with many scans. The alignment errors can be reduced by using known targets and determining their coordinates by conventional surveying techniques. In any case, aligning the scans without a topographic control network does not seem convenient, not even for small buildings. The control-network adjustment accuracy is undoubtedly related to the scan alignment accuracy. The workflow focused on the following steps and issues:

- Methods for checking the scan alignment;
- Possibility of checking and refining the scan direct orientation by scan-to-scan alignment;
- Accuracy checks for direct scan alignment for the purposes of surveying the architectural context.

*Scan Alignment Check*

The outcomes of the cloud-to-cloud alignment encompass the residual error and the percentage of overlap, which can be evaluated against predetermined tolerances. However, such a tool is not available in direct topographic scan orientation. The project cloud can be segmented into slices, and the scan alignment can be managed by visually inspecting each slice. The visual inspection suffices for standard applications in conjunction with an accuracy check of the control-network adjustment (Figure 3).

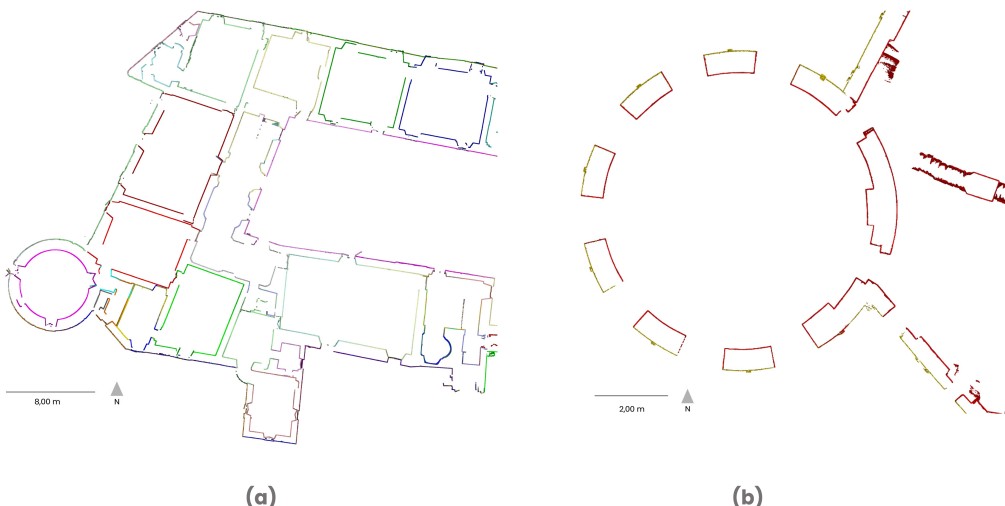

**Figure 3.** Visual alignment check on a horizontal slice (**a**). Visual indoor/outdoor point cloud alignment check on a horizontal slice. Detail of the southwest circular tower (**b**). Each scan is in a different color.

Another method for assessing misalignments is the cloud-to-cloud analysis, which is typically applied outdoors when substantial overlap between scans exists. This tool is incorporated in CloudCompare [16]. It has been used with the following parameters:

- Octree level: Auto;
- Maximum distance: 0.02 m (larger distances are attributed to non-overlapping cloud areas);
- Local model: quadric (in order to reduce the noise effect);
- Points (kNN): 6.

The outdoor survey comprised 20 scans, and eight pairs of well-overlapping clouds were analyzed, both prior to and following an ICP-enhanced alignment. Cloud-to-cloud distances were subdivided into classes, computing for each class the statistics:

$$f_i = \frac{n_i}{\max(n_i)}$$

$$F_i = \frac{\sum_{j=1}^{i} n_j}{N}$$

where $f_i$ and $F_i$ are the relative frequency and cumulative frequency, $n_i$ is the number of points inside a class and $N$ is the total number of points. Note that the relative frequency $f_i$ is normalized by the maximum number of points inside a class. The function maximum is always 1 to quickly compare the maximum value position with other point cloud pairs, as in Figure 4.

The worst alignment was detected in the couple C05-A02. The computed statistic $f_i$ presents a maximum around the distance of 2.5 mm, representing the misalignment mean error, a misalignment that, from a practical point of view, is negligible, among other things, cannot be noticed in a visual inspection. Note that:

- The horizontal coordinates of C05 were, however, determined with an accuracy of 4 mm with respect to A02, and the vertical coordinate with an accuracy of 2 mm;
- Station C05 has been oriented on B01 and not on A02, so the two stations are not directly connected;
- A station orientation error can lead to cloud-to-cloud misalignment, even if the coordinates of the instrumental origin are known without errors.

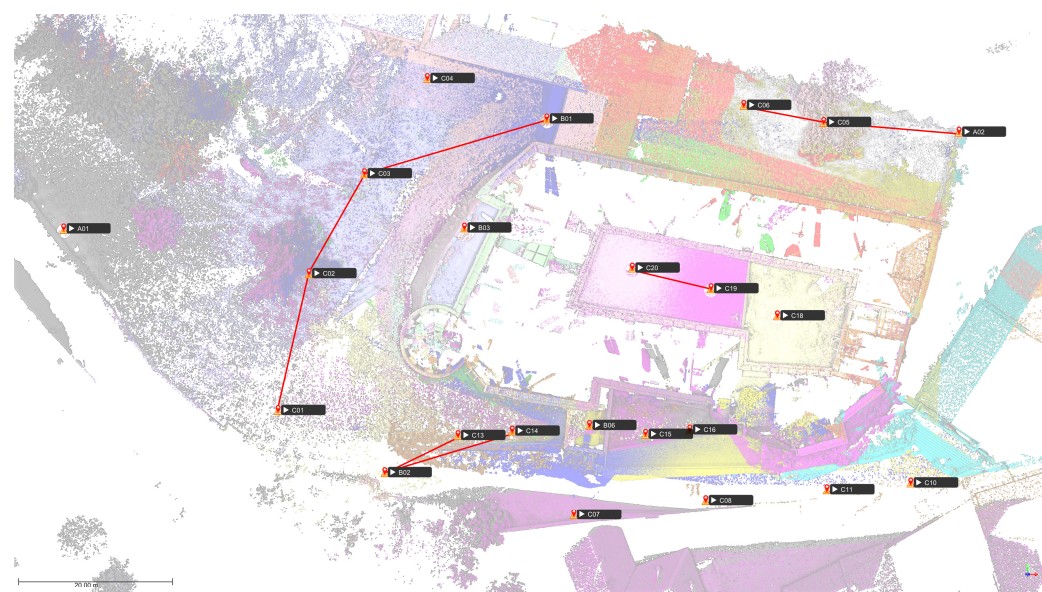

**Figure 4.** 3D scans outside the castle. The connections in red highlight the eight scan couple alignments verified in this work.

In Figure 5, the statistics $f_i$ and $F_i$ before alignment refinement are represented in red, and after in green.

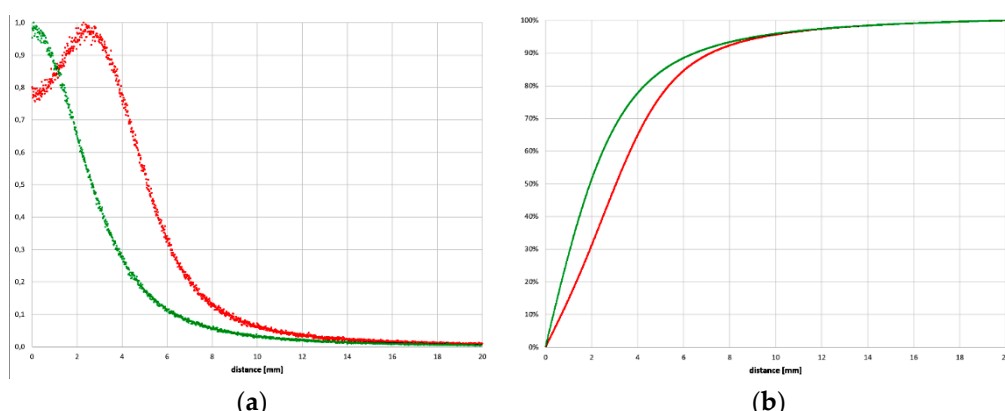

(**a**)          (**b**)

**Figure 5.** (**a**) $f_i$ statistic. (**b**) $F_i$ statistic. Red: Before alignment refinement. Green: After ICP alignment refinement.

The statistic $F_i$ shows that 95% of the cloud-to-cloud distances are under 10 mm. It also shows that before the alignment refinement, only 65% were under 4 mm, and after, it increased to 78%. Plotting the three components X, Y, and Z, of the cloud-to-cloud distance before the alignment, a shift can be observed around 2 mm in Y, while X and Z have zero mean. The shifts estimated by ICP are −1.1, 16.5, and −0.3 mm, according to the shifts observed in the cloud-to-cloud distances (Figure 6).

Similar statistics were computed over the ten examined scan couples before and after the alignment refinement. To understand the impact of the ICP refinement, here the differences in the cloud-to-cloud distance, both in $f_i$ and mostly in $F_i$: $f_i$ maximum shifts to zero distance, while $F_i$ must necessarily increase on the left side of the curve to have an alignment refinement (Figures 7–9).

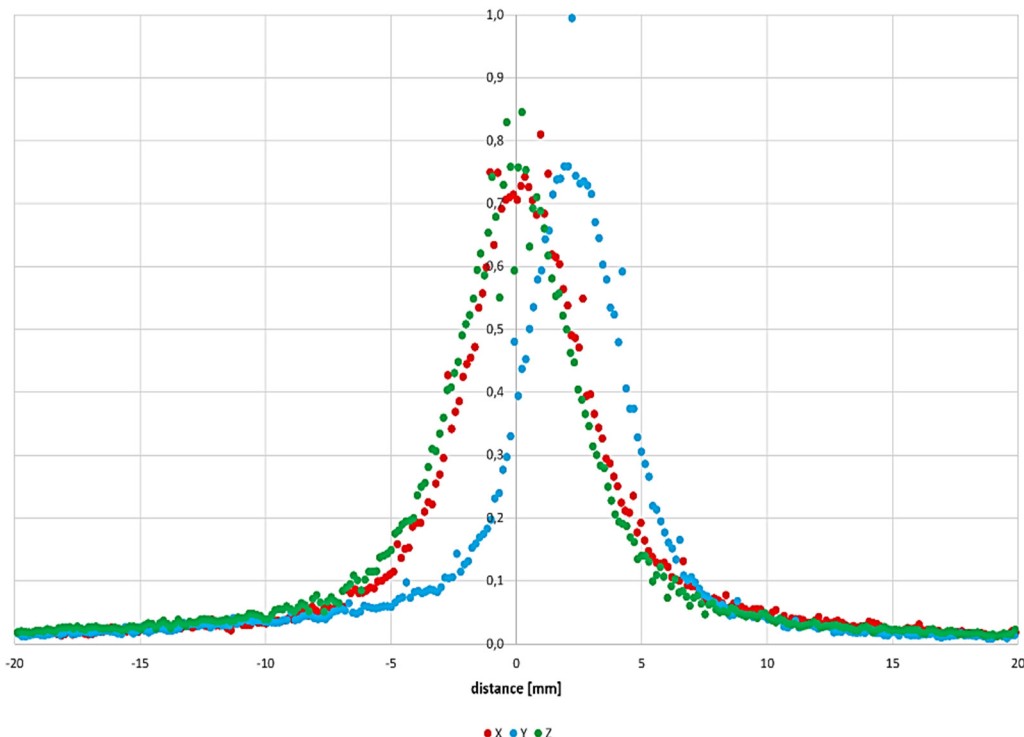

**Figure 6.** Cloud-to-cloud distance components.

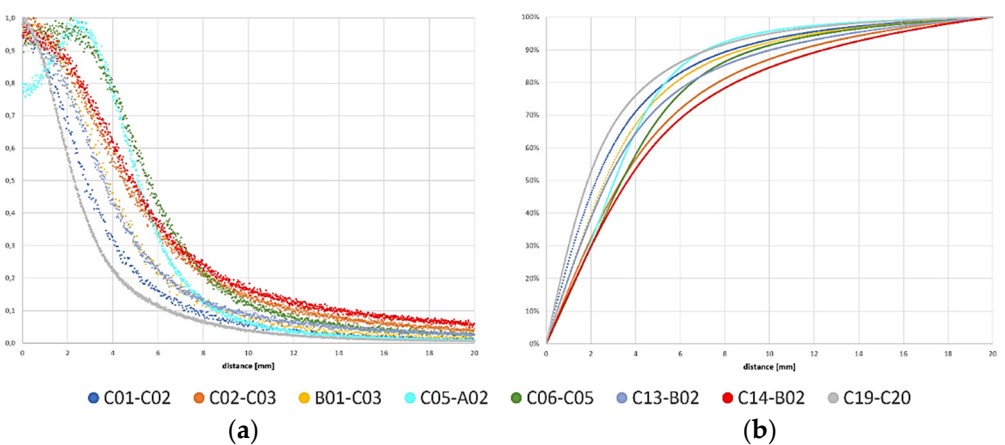

**Figure 7.** Statistics of the scan couples before the alignment refinement. (**a**) $f_i$, (**b**) $F_i$.

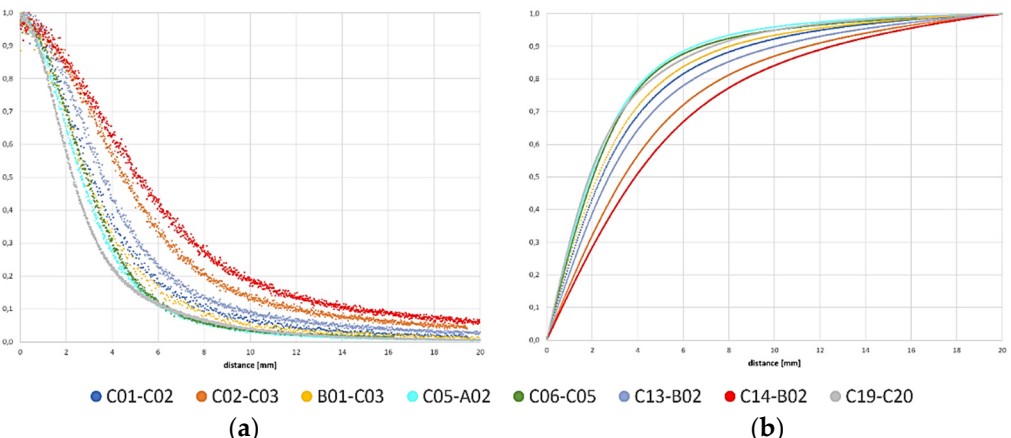

**Figure 8.** Statistics of the scan couples after the alignment refinement. (**a**) $f_i$, (**b**) $F_i$.

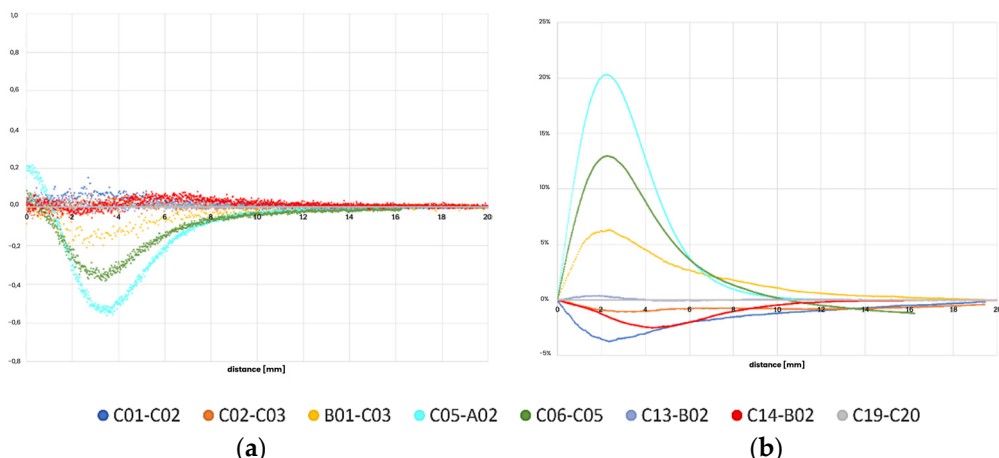

C01-C02    C02-C03    B01-C03    C05-A02    C06-C05    C13-B02    C14-B02    C19-C20

(**a**)                                                                                      (**b**)

**Figure 9.** Differences in the statistics of the scan couples after and before the alignment refinement. (**a**) $f_i$ after ICP minus $f_i$ before and (**b**) $F_i$ after ICP minus $F_i$ before.

Surprisingly, there was no improvement in alignment. This only happened for the couples C05-A02, C06-C05, and B01-C03. There is practically no variation for couples C13-B02 and C19-C20, while a slight worsening was seen for couples C02-C03, C14-B02, and C01-C02. So, cloud-to-cloud alignment can improve scan orientation in case of misalignments. Still, its application is not always convenient when an excellent topographic scan orientation is performed.

Predominantly in indoor surveying scenarios, the registration from one scan to another proves unfeasible due to the exiguous or non-existent overlapping regions. Scan stations are typically situated near the room's center and are interconnected via a topographic station, commonly located in the passageway that links the rooms. The existence of frames, doors, and windows introduces complexities; when left open, they hinder the visibility of wall sections, and when closed, they preclude the complete scanning of neighboring rooms (Figure 10).

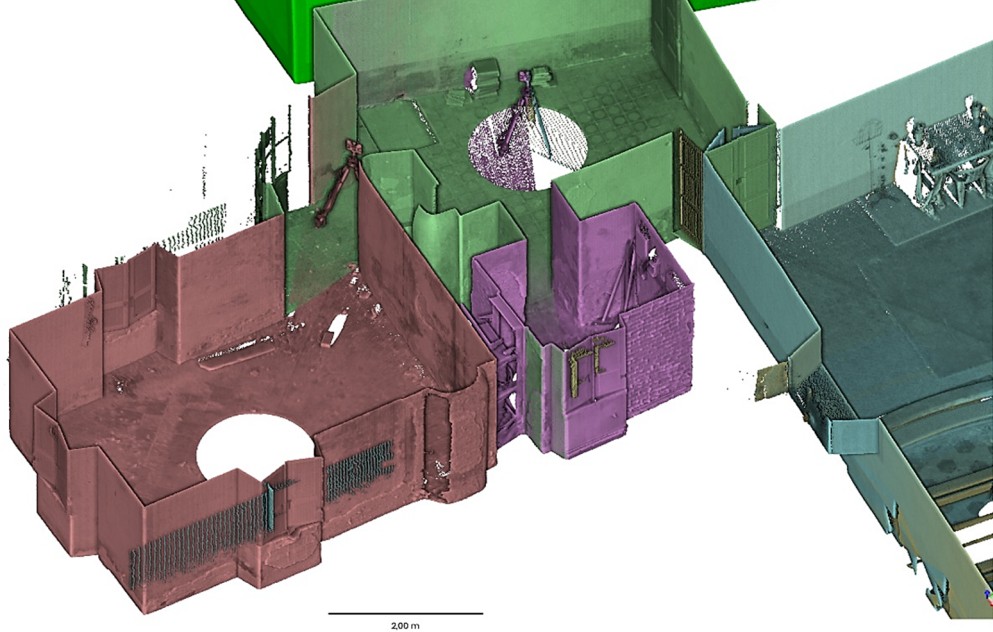

**Figure 10.** View of a portion of the indoor survey of Frinco Castle. Each scan is in a different color. A minimal overlap between adjacent scans can be observed.

Another common situation is the survey of buildings with multiple overlapping floors. In these instances, the intrados and extrados of the vaults were surveyed to gauge their thickness (Figure 11). The linkage of the scans is exclusively facilitated by topographic measurements, occasionally necessitating extensive traverses to establish a connection with the external control network. Logistically, linking the indoor survey to the external control network can be lengthy, often requiring additional personnel to relocate tripods and prisms outdoors. To streamline this process, the team planned to anticipate the need for outdoor relocation of tripods during the scanning phase, maintaining clear and continuous communication channels between indoors and outdoors.

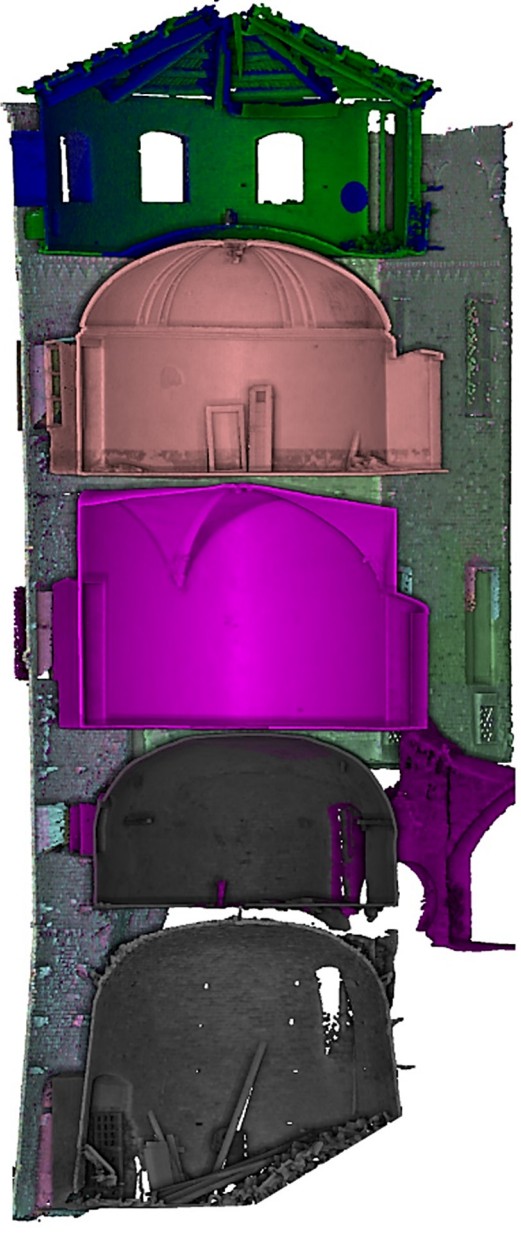

6,00 m

**Figure 11.** View of a portion of the outdoor and indoor survey of Frinco Castle. Vertical section of the circular tower at the southwest corner of the castle. Each scan is in a different color.

Common features and patches can be used for indoor scan alignment, or the control network adjustment can be checked in other cases. To maximize the possibility of having

common entities between indoor and outdoor scans, it is preferable not to set a limit on the scanning distance during the acquisition or when importing the scan data. An estimate of the cloud-to-cloud distance will be helpful for understanding if common areas are present among scans large enough to provide a meaningful statistic. The scanner can acquire external areas through an open window, for example.

Then, the following control procedure, based on the use of shared features between indoor and outdoor scans, was adopted to check the alignment of a single indoor scan (Figure 12):

1.  Exclude the indoor portion from the indoor scan. When combining indoor and outdoor scans, alignment and registration become critical. Excluding the indoor part simplifies this process, reducing the number of variables to consider and focusing on the relevant data.
2.  Restrict the outdoor point cloud to the indoor bounding box.
3.  Approximatively restrict the outdoor point cloud to the overlapping area, if any.
4.  Compute the indoor/outdoor cloud-to-cloud distance, taking the indoor one as the reference cloud.
5.  The points with a cloud-to-cloud distance larger than 0.05 m are excluded from the outdoor cloud to extract the common patch.
6.  Compute the indoor/outdoor cloud-to-cloud distance, taking the outdoor one as the reference cloud.
7.  Filter the outdoor cloud by distance scalar field, selecting the range (0.00–0.02 m) to be used for the statistical analysis.
8.  Compute the cloud-to-cloud distance distribution and the $f_i$ and $F_i$ statistics.

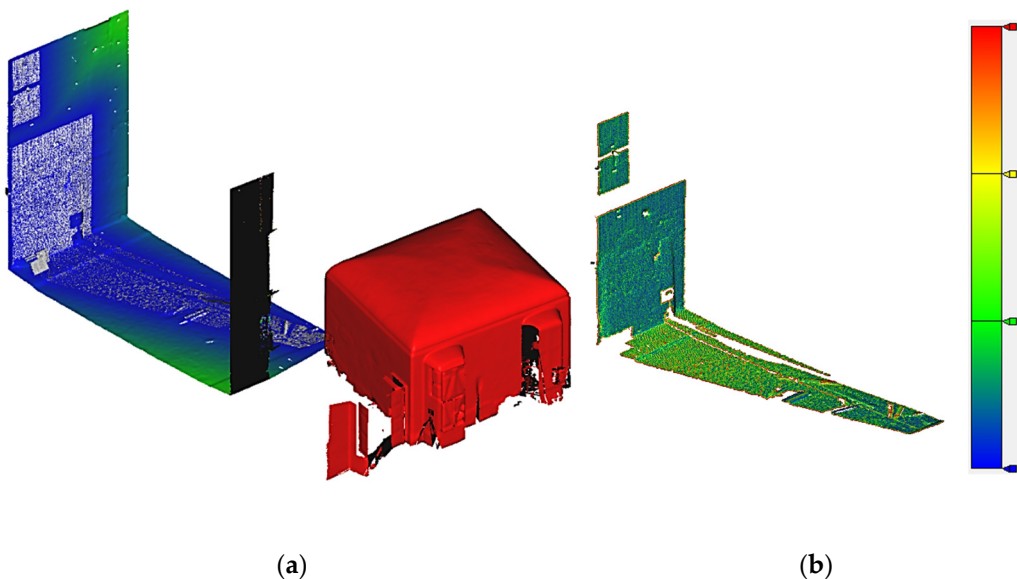

(**a**)                                                                                  (**b**)

**Figure 12.** Alignment check of an indoor scan, where the outdoor part overlaps with an outdoor scan. (**a**) The indoor part, in Red, is eliminated, and the cloud-to-cloud distance is computed on the overlapping outdoor part (steps 1–2–3) (**b**) Cloud-to-cloud distance of the overlapping part (steps 4–5). The Blue–Green–Yellow–Red color scale represents the distances from the minimum (Blue) to the maximum value (Red).

If the statistic $f_i$, representing a measure of scan alignment quality, shows a non-zero maximum different from zero, a misalignment to the external point cloud was registered. In this case, it is necessary to assess the severity of the misalignment. Factors to consider include the scale tolerance and evaluating whether the misalignment is within an acceptable scale tolerance. Small discrepancies may be tolerable because minor misalignments may not significantly affect the final deliverables. More significant deviations require correction using algorithms like ICP to perform scan alignment based on point-to-point

correspondence iteratively. Note that ICP alignment can be performed if the overlapping parts are sufficiently significant both in the horizontal and the vertical components.

Here is an example of the indoor/outdoor scans alignment check on scans C131, C134, C133, and C135, which share parts of 266, 141, 181, and 166 points in common with the outdoor point cloud: The common parts are not very extensive, but the example is quite demonstrative of the applied control method. As seen in the following graph (Figure 13), misalignments of up to 2 mm can be detected, which are within the scale tolerances of the architectural survey.

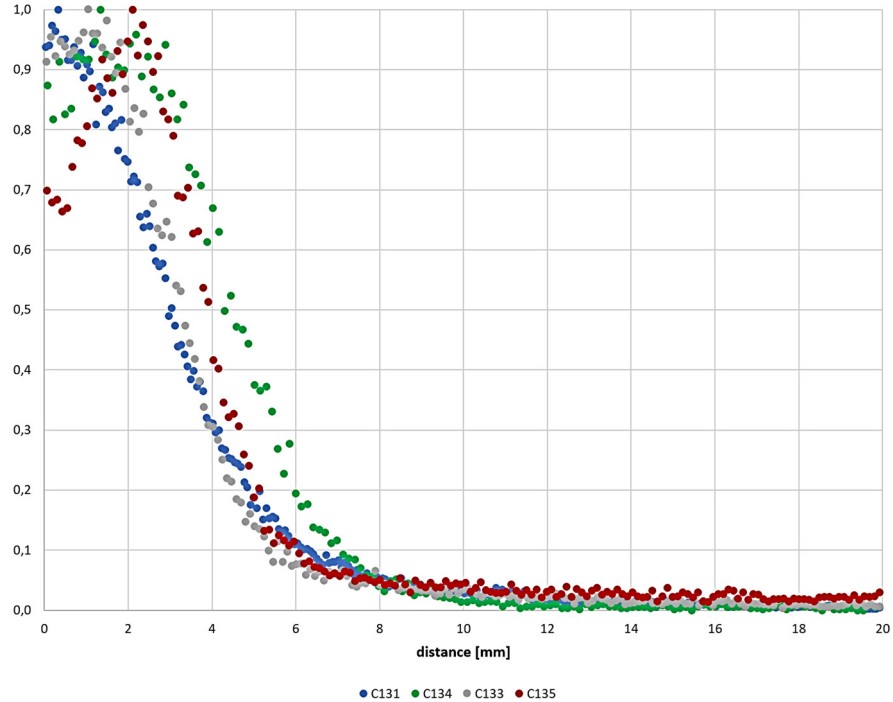

**Figure 13.** $f_i$ statistic of the indoor/outdoor scans alignment check. Cloud-to-cloud distances are computed with respect to the outdoor point cloud. The series maximum represents the detected 3D shift.

## 4. Data Fusion

### 4.1. Terrestrial Scans

Combining different scans of the same object, it is common practice to accept all the data from all the scans to be combined, to select only the points from each scan up to a maximum scanning distance, or to make a manual selection. It is clear, however, that it would be convenient to have a tool capable of selecting the points with the highest scanning quality from the different scans in the overlapping areas.

Before establishing selection criteria grounded in scanning quality, it is crucial to acknowledge that scanning precision is influenced by several factors. These include the scanning distance and the surface's reflectivity, but most notably, the angle of incidence at which the laser beam intersects with the surface of the object being scanned [17,18]. The angle at which the laser beam strikes the surface determines the quantity of reflected light captured by the scanner. Consequently, the quality of the data gathered and its underlying rationale are fundamentally geometric:

- Specular reflection: The reflectivity of a surface can change depending on the angle of incidence of the laser beam. For example, when the laser beam hits the surface at an acute angle, there is a greater probability that the beam will not be reflected toward the scanner but in a different direction, leading to a lack of data from that particular surface area.

- Geometric distortion: Sharp angles of incidence can cause geometric distortion because the laser spot is larger than when the beam hits the surface perpendicularly.

The angle of incidence has been studied primarily for its effect on the intensity value. However, it was exploited to filter out points measured with an angle of incidence more significant than a threshold value. Calculating the laser's incidence angle with respect to the normal to the surface is very simple. For a terrestrial scan with scan origin $X_0$, $Y_0$ and $Z_0$, the local coordinates were used:

$$\vec{p}_i = \begin{pmatrix} x_i \\ y_i \\ z_i \end{pmatrix} = \begin{pmatrix} X_i - X_0 \\ Y_i - Y_0 \\ Z_i - Z_0 \end{pmatrix}$$

To compute for each point the distance from the scan origin:

$$D_i = \sqrt{x_i^2 + y_i^2 + z_i^2}$$

and the components of the scan origin direction:

$$\vec{d}_i = -\vec{p}_i / D_i$$

The unit vector $\vec{n}_i$ normal to the surface is computed in CloudCompare, setting the scan origin $(X_0, Y_0, Z_0)$ as preferred orientation. The incidence angle $\delta_i$, that is, the angle between the vectors $\vec{d}_i$ and $\vec{n}_i$, can be computed as:

$$\delta_i = \cos^{-1}\left(\vec{d}_i \cdot \vec{n}_i\right)$$

The angle $\delta_i$ ranges from $0°$, when the scan laser beam is normal to the surface, to $90°$, when the scan beam is tangent to the surface. As demonstrated in [8], the scan standard error increases exponentially at incidence angles larger than about $60°$–$70°$.

The angle $\delta_i$ must be considered by combining two scans. Observe the two scans, A and B, of the same building (Figures 14–18). They have non-overlapping areas that must be retained. It is in the overlapping area where a quality-based selection can be performed. The following steps were designed:

(1) Select and retain the non-overlapping point cloud areas from A and B;
(2) On the overlapping areas:

    (a) Compute the incidence angle δ for the two scan A and B;
    (b) Choose from A or B the points that have incidence angle lower than a fixed threshold (i.e., 70°).

(3) Merge with non-overlapping point cloud areas from A and B.

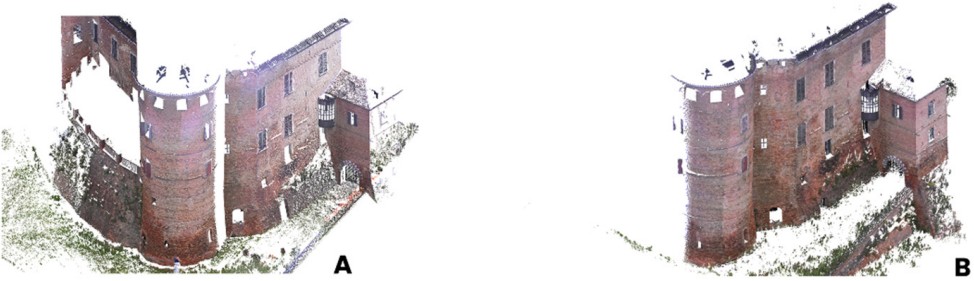

**Figure 14.** Point clouds of the original scans: focus on the Scan (**A**,**B**).

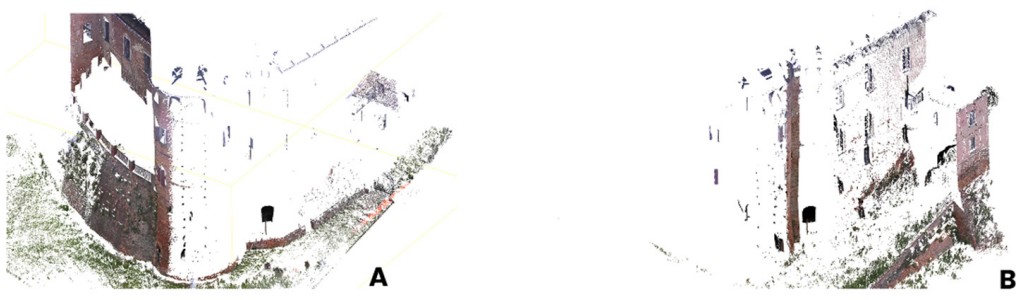

**Figure 15.** Non common parts of Scan (**A**,**B**).

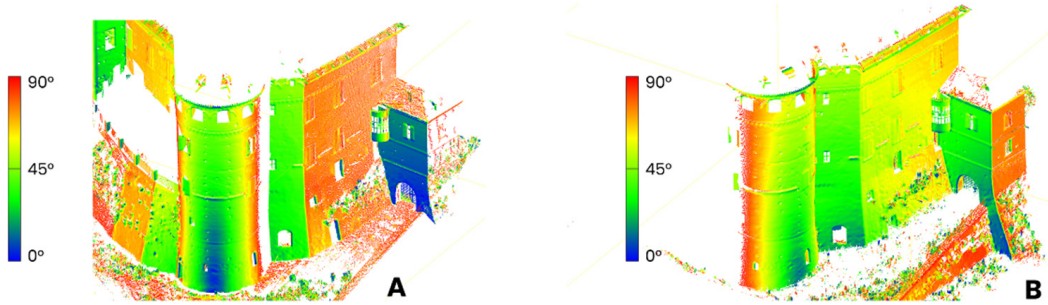

**Figure 16.** Scan (**A**,**B**): Incidence angle δ (angle between the scan direction and the normal to the object surface).

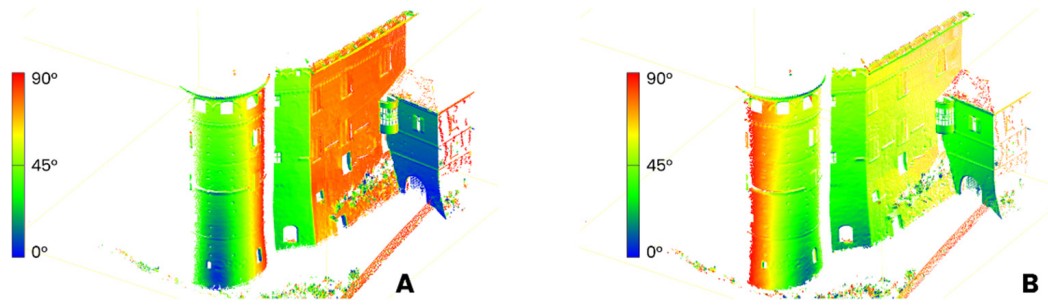

**Figure 17.** Scan (**A**,**B**): Incidence angle δ on the overlapping area.

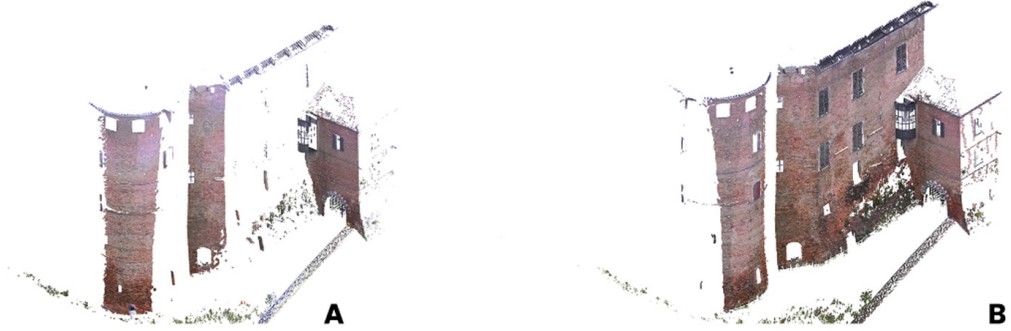

**Figure 18.** Extracted point clouds from Scan (**A**,**B**).

The result is a merged cloud of the A and B scans, cleaned by noisy points in the areas scanned at high incidence angles (Figure 19).

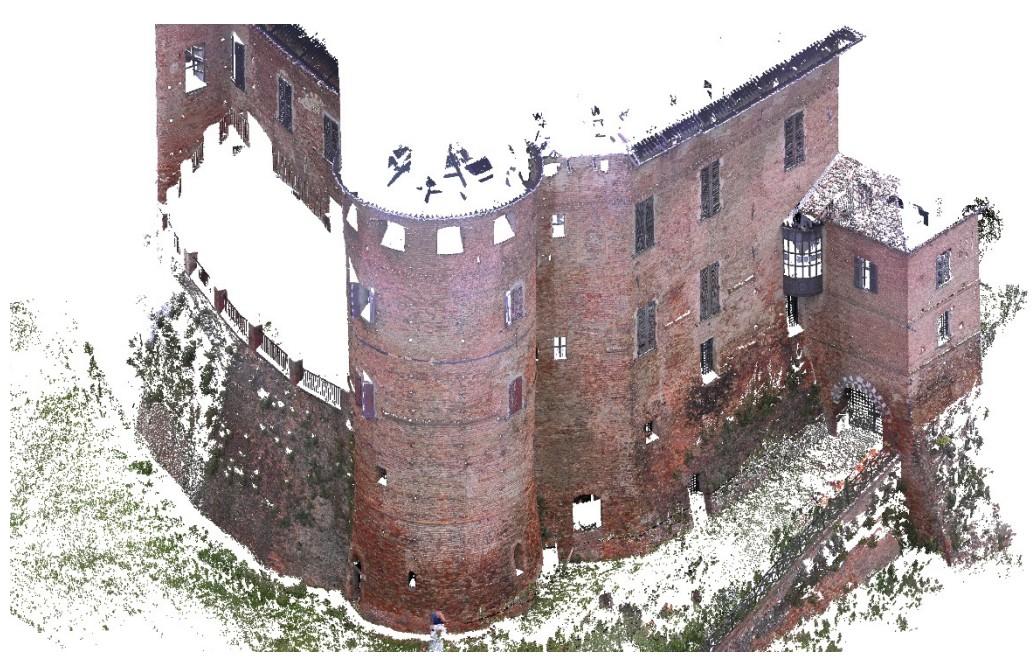

**Figure 19.** Fused point clouds related to different scans.

*4.2. Terrestrial and Airborne Scans*

Airborne point clouds can be acquired by photogrammetry or by scanning. At Frinco Castle, the metric survey was carried out both by DJI (Shenzhen, China) L1 LiDAR and P1 camera, and the data comparison has already been discussed by the authors in [6]. The photogrammetric process was operated by using Agisoft Metashape (Figure 20) [19].

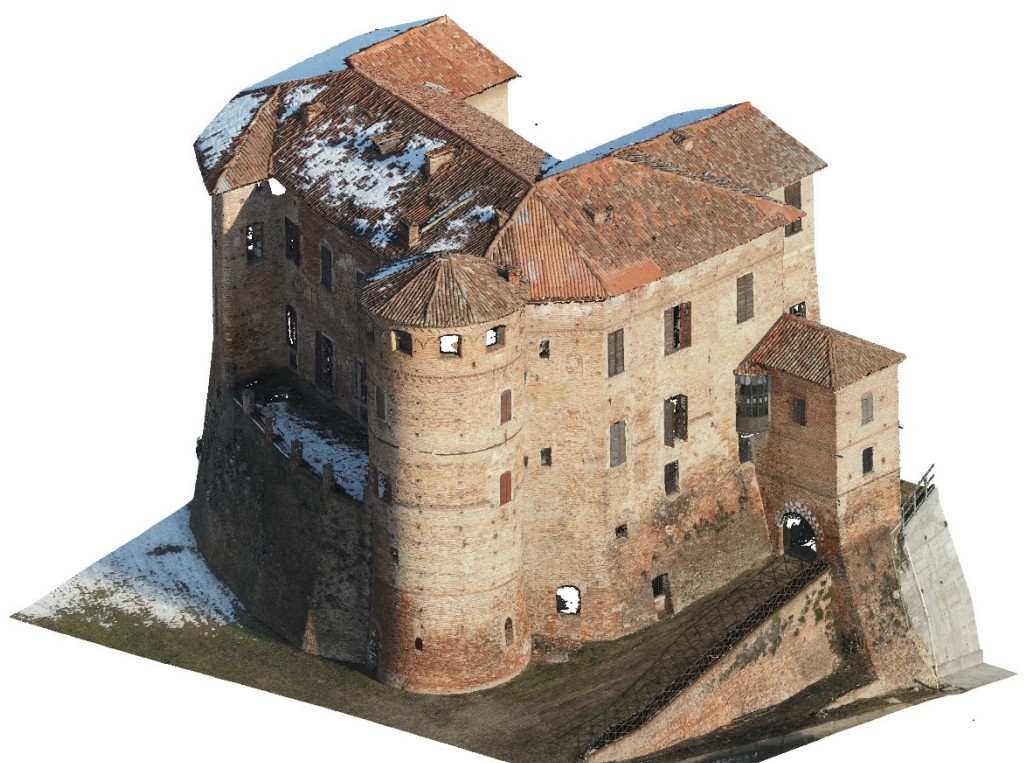

**Figure 20.** The photogrammetric point cloud acquired by the DJI P1 camera.

The merging process of ground and airborne point clouds is simple, assuming that the ground point clouds are always more accurate and less noisy than the airborne ones.

Using the cloud-to-cloud comparison in CloudCompare, it is possible to cancel out the airborne point cloud in the voxels occupied by ground scans (Figure 21). The size of the voxels depends on the scan's sampling distance, and in architectural survey, 1–2 cm is a good value. The procedure is much simpler and faster than the manual segmentation of the airborne point cloud, but alignment errors can lead to undesired point selection (Figure 22). Point cloud alignment can be easily controlled through the cloud-to-cloud distance on the overlapping area. In this way, the combined point cloud was obtained (Figures 23 and 24).

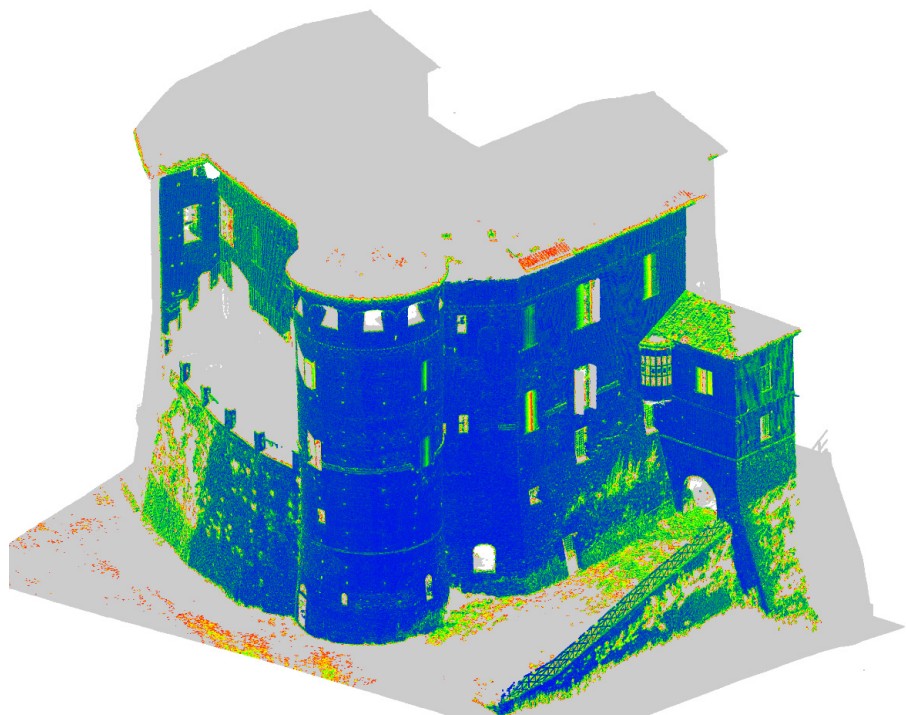

**Figure 21.** Cloud-to-cloud distances computed by CloudCompare on the airborne/terrestrial scan couple. The grey areas belong to the airborne scan and will be selected for data fusion.

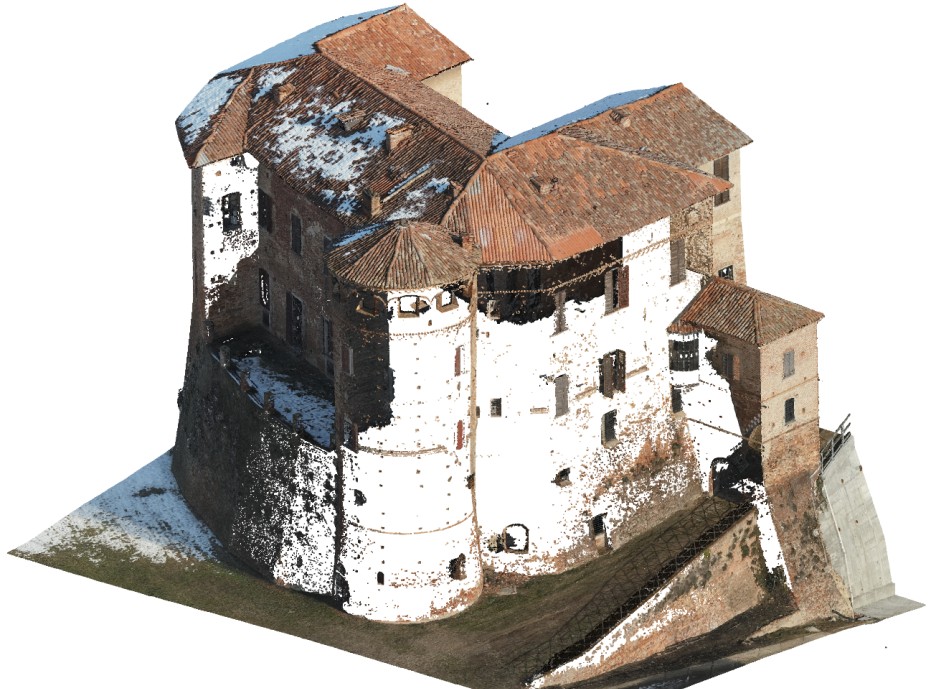

**Figure 22.** Photogrammetric point cloud of the selected points.

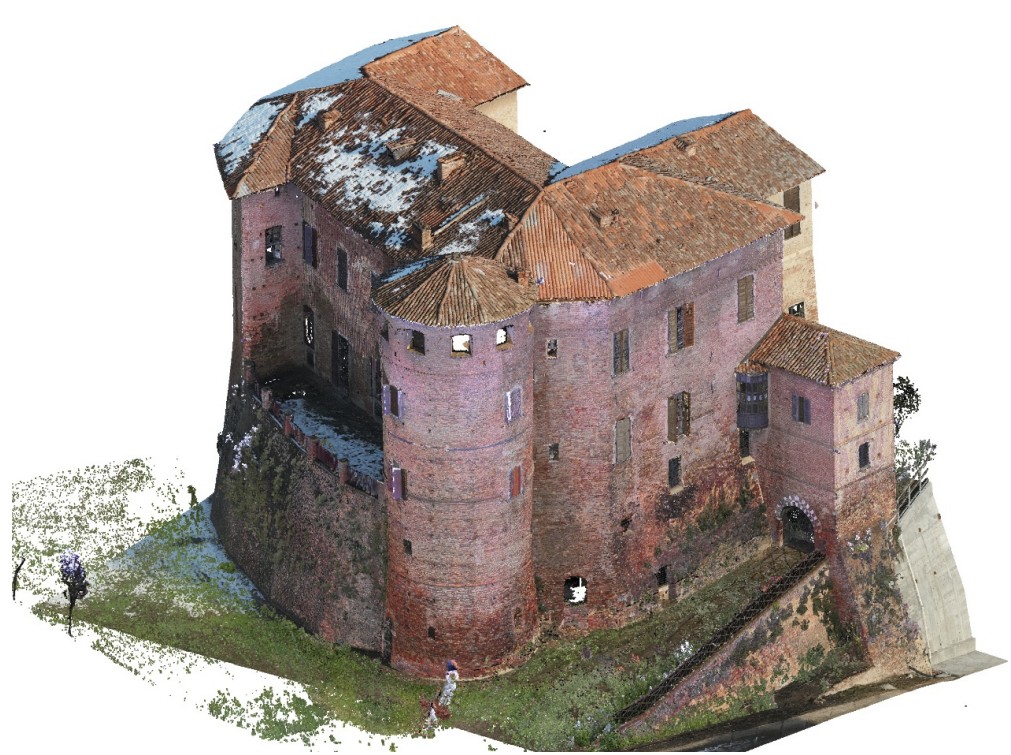

**Figure 23.** Airborne/terrestrial combined point cloud.

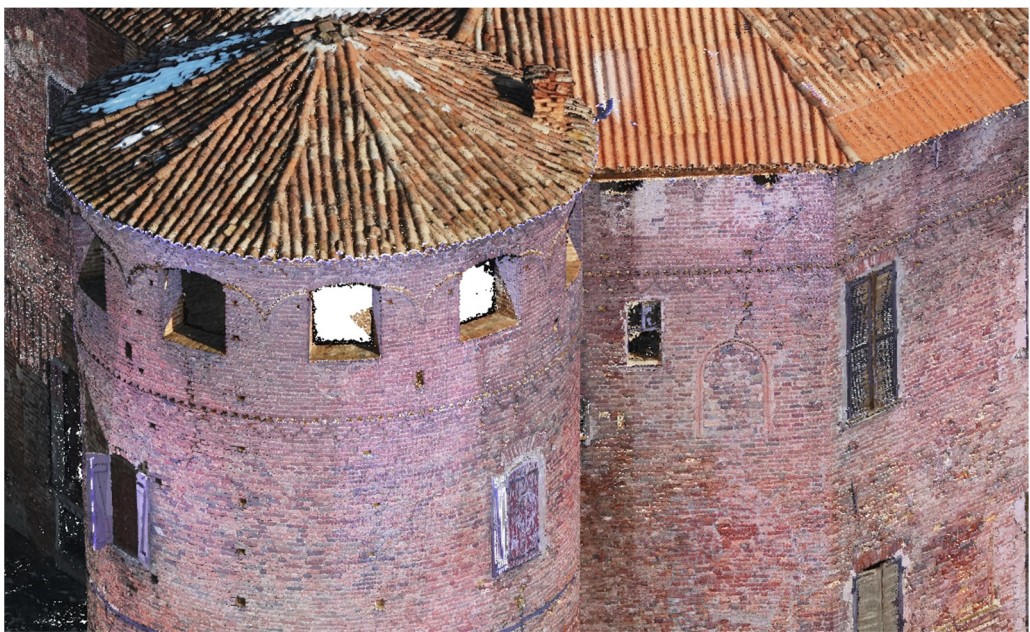

**Figure 24.** Detail of airborne/terrestrial combined point clouds.

## 5. 3D Modeling

Point clouds resulting from the multi-sensor data fusion were the structural backbone of the 3D modeling phase. In this regard, reality-based modeling was performed to achieve a detailed polygonal mesh with RGB values, starting from data fused: terrestrial and aerial acquisition, indoor and outdoor. Then, a digital twin (from a graphical and metric point of view) of the castle was obtained. In addition to the polygonal mesh modeling, two different approaches have been operated: parametric modeling of structural elements of roof parts (wooden beams); NURBS modeling of the entire castle.

RGB values are not calibrated. All the terrestrial and aerial photography was acquired on cloudy days to produce uniform and diffuse illumination. The deliberate use of cloudy days ensures that the RGB values in the acquired images are not affected by extreme lighting variations, contributing to better data quality.

### 5.1. Parametric Design

TLS 3D data was exploited for creating parametric components (structural features) related to the internal parts of the castle, specifically those related to the collapsed area (where the aerial data were also used). The parametric design, using Trimble Real Work software (v.12.0) [4], included the structural parts of the roof, such as wooden beams (Figure 25). This process was carried out through cloud-based adaption: parametric surfaces and geometries were dynamically adapted to 3D point clouds. The parametric modeling process was relatively quick (approximately 2 min for beam) due to point cloud simplification and segmentation before the geometric adaptation: in this way, dynamic cubes were adapted (length–height–thickness values) to segmented point cloud of beams.

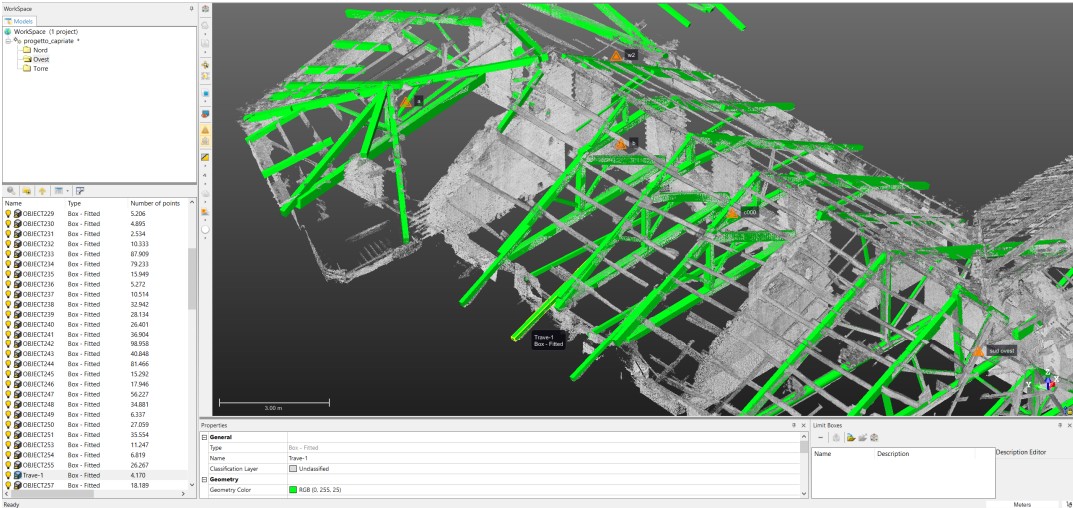

**Figure 25.** Parametric design related to wooden beams of the rooftop: geometries adapted to point clouds.

This modeling choice has been made especially to have a clear and simplified overview of the structural conformation of the internal rooftop, which is fundamental for creating detailed 2D drawings of indoor situations essential for future reconstruction.

### 5.2. NURBS Modeling

Metric data related to point clouds and wires from planar sections were the focus of the NURBS (Non-Uniform Rational Basis-Splines) modeling, detailed explained in previous work [15]. A simplified and lightweight model was developed that maps the features of masonries and the architectural stratigraphy of the castle. In fact, the complexity of the castle is given not only by the architectural components but also by the stratigraphic layers of masonries. In fact, the project presented here required extensive free-form modeling that was useful for different purposes.

By using Rhinoceros software (v.7.0) [20], curves and surfaces were interpolated and adapted to wired profiles resulting from point clouds: the complete reconstruction was assigned, combining *extrusions*, *loft*, *sweep*, and *patch* commands.

Firstly, the current situation of the southern front relative to the collapsed area was modeled by reconstructing the collapsed portion (volume) from historical photographs and actual measurements from the integrated survey and then simulating the structural collapse through the use of cutting planes (Figure 26). This modeling phase was fundamental for formulating a new hypothesis of reconstruction (according to local authorities).

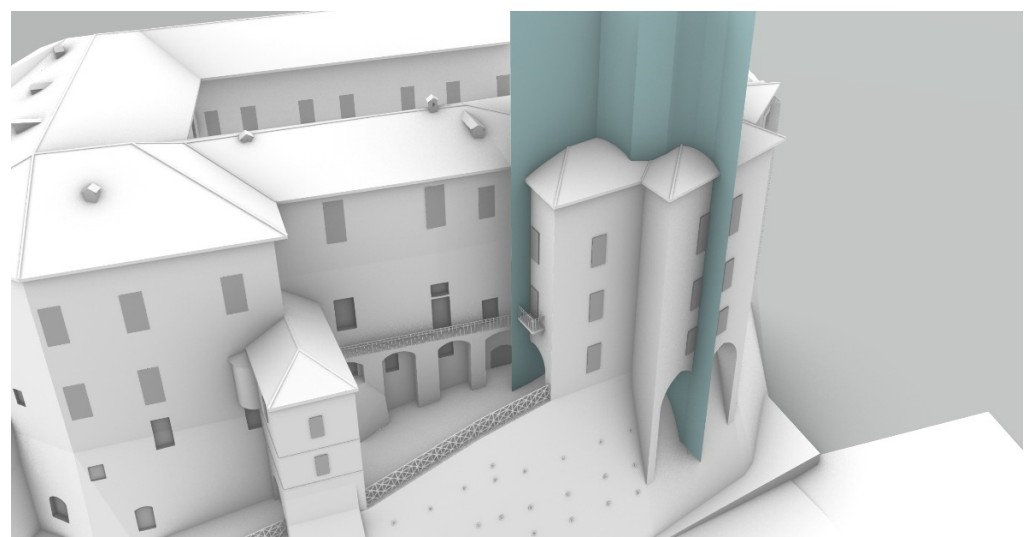

**Figure 26.** NURBS model of Frinco Castle: Simulation of the structural collapse through the use of cutting planes.

The second modeling step concerned the complete free-form reconstruction of the defensive building, including the collapsed area and the internal vaulted rooms. Modeling the entire castle enabled the possibility of a dynamic and lightweight 3D representation on which mapping detailed HD orthophotos and the stratigraphic analysis performed on masonries (Figure 27). Reading stratigraphic units, which are construction site layers (including additions and modifications), was fundamental to formulating a chronological interpretation and then architectural evolution of the castle. The masonry stratigraphy and its interpretation were discussed in a previous work [15]. The multi-sensor approach and data fusion revealed the necessity for the complete 3D documentation of the external and internal covering elements of the castle, especially related to the collapsed area.

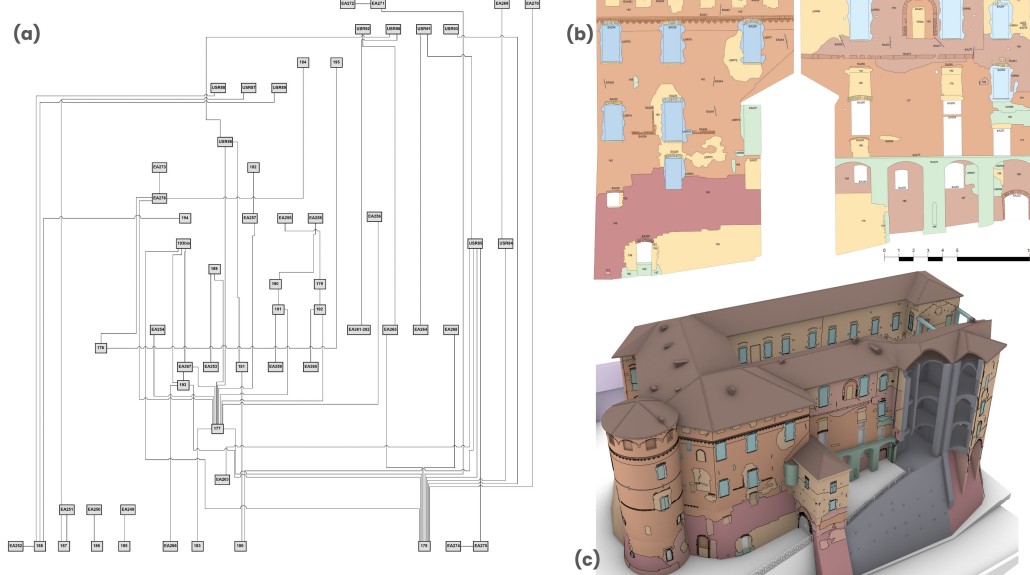

**Figure 27.** From stratigraphic analysis to 2D drawings and 3D mapping NURBS model. This figure shows a focus on the south front and the collapsed area of the castle: (**a**) stratigraphic diagram (matrix); (**b**) historical interpretation of stratigraphy; (**c**) stratigraphic units mapped into the NURBS model depending on historical periods.

### 6. Discussion

The suggested approach for verifying scan alignment, applicable to outdoor and indoor scenarios, can be employed for random checks or to identify misalignments that necessitate refinement via ICP. Our case study identified more pronounced misalignments of a few millimeters (2–4 mm), and in this case, ICP refinement is not mandatory.

The Trimble SX10 surveying solution has proven its reliability and precision across all tested conditions for outdoor and indoor architectural surveys. A robust and accurate control network guarantees the direct alignment of scans with an accuracy sufficient for creating architectural drawings and other deliverables typically required in architectural surveys. The primary limitation related to a TLS is the scan rate. However, the system's ability to avoid post-survey scan registration partially offsets the extended scanning duration. The benefit of minimal noise in scan point clouds was evaluated and further reduced by eliminating points scanned at a high incidence angle.

The principal advantage of direct scan referencing by SX10 lies in the indoor scan alignment, mainly when shared features or targets can be challenging to locate or place.

The quality-based selection criteria used to merge different scans can be easily applied by processing in CloudCompare. The scan distance and incidence angle can be computed using Scalar Fields Arithmetic, while to compute the Normal Vector, the standard algorithm implemented in CloudCompare was used, selecting the scan origin as the preferred orientation (previously shifting the scan origin to the coordinate origin). The cloud-to-cloud distance calculation and the command "filter points by value" were exploited to select a subset of the point cloud. The selection of the maximum threshold value for the incidence angle depends on many factors, at first on the availability of redundant data in overlapping scan regions, then on the quality of the data and their level of noise, which are functions of the scanned materials reflectance, of the distance and the scanner quality.

### 7. Conclusions

This work and manuscript focused on the multi-sensor metric survey of Frinco Castle. A general overview of the techniques used to generate point clouds and the alignment check, segmentation, and data fusion was provided. Outdoor and indoor surveying followed two workflows, similar to other projects [21,22], where only outdoor surveying needed data fusion.

Frinco Castle required careful and ad-hoc survey planning to acquire all the building components. Data integration and fusion were the only possible ways to have a complete metric overview of the castle: multi-sensor surveys, performed with aerial and terrestrial data, demonstrated its reliability for complex architectural contests [23,24].

Integrated and fused point cloud data was the starting point for preparing the architectural parametric and NURBS modeling, designed for achieving different outputs: 2D detailed drawings (floorplans, sections) and NURBS dynamic model.

The 3D survey and NURBS modeling were also designed to help the stratigraphic investigation of external masonries, a needed workflow for building archaeology projects. This information was essential for producing highly detailed orthophotos that were used to map the stratigraphic layers of architectural fronts and, subsequently, to comprehend how the defensive building's architecture evolved [15,25].

Lately, the partial parametric modeling initially planned for having a 2D drawing simplified view of the internal areas (especially the roof) could prove to be fundamental for designing an entire parametric architectural model of the castle for handling metric and semantic data in a dynamic Historic Building Information Modeling (HBIM) environment. The acquired metric data and related digital outputs could be the basis for building up an informative model of the castle. This proposal is also under consideration to have a standard and intelligent platform for the upcoming restoration phase.

The HBIM project of the castle should be designed to preserve metric and semantic data, especially related to the historical interpretation and structural information of this fragile building. Being a 3D database and data-exchange platform, the HBIM model would

enable the implementation of the castle's collected historical data and synchronic collaborations for further analyses [26–28]. Professional figures involved in the restoration project could analyze and exchange sensitive data before hands-up operations. This proposal will also be designed through a tailor-made Common Data Environment (CDE), which enables users and collaborators to access, read, edit, and query historical information inside the parametric informative model. This possibility breaks data accessibility barriers and eases forthcoming restoration processes.

**Author Contributions:** Conceptualization, M.R. and F.D.; methodology, M.R. and F.D.; software, F.D.; validation, M.R. and F.D.; formal analysis, M.R. and F.D.; investigation, M.R. and F.D.; resources, M.R. and F.D.; data curation, M.R. and F.D.; writing—original draft preparation, M.R. and F.D.; writing—review and editing, M.R. and F.D.; visualization, M.R. and F.D.; supervision, M.R. and F.D.; project administration, M.R.; funding acquisition, M.R. All authors have read and agreed to the published version of the manuscript.

**Funding:** This research was funded by Fondazione Cassa di Risparmio di Asti.

**Data Availability Statement:** Data are contained within the article.

**Acknowledgments:** The authors would like to acknowledge the Frinco municipality for granting and supporting topographic and aerial investigations. Special thanks go to Giorgio Viazzo and Chiara Viazzo for contributing and supporting the terrestrial acquisition phase and for the production of charts and elevations.

**Conflicts of Interest:** The authors declare no conflicts of interest.

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
