# Peer review of "Multi-Sensor 3D Survey: Aerial and Terrestrial Data Fusion and 3D Modeling Applied to a Complex Historic Architecture at Risk"

_drones, doi:10.3390/drones8040162_

Round 1

Reviewer 1 Report

Comments and Suggestions for Authors

- The introduction needs improvement to clearly outline the paper's objectives.

- Please provide more elaboration on the formulas used, particularly lines 182 and 183.

- In line 325, please confirm if the correct symbol is being used between 60-70.

- Authors should provide additional explanation for Figure 25.

Comments on the Quality of English Language

The entire manuscript should be carefully reviewed once more for a final polish of the English language.

Author Response

Dear Reviewer,

The authors would like to thank You for your interest in this manuscript and for your effort in this review. In this revised version we applied all the suggestions You pointed out:

  • Introduction chapter was implemented including the goal of the project
  • The statistics fi and Fi (relative frequency and cumulative frequency) have been defined explicitly
  • the use of the symbol ÷ is correct
  • Figure 25 (now figure 27) enriched with additional information in the text and caption of the figure
  • english has been reviewed

The authors are grateful to You for your important suggestions and corrections.

Thanks again.

Reviewer 2 Report

Comments and Suggestions for Authors

Please see comments in the file. 

Comments on the Quality of English Language

Improve the informal descriptions, and the many unclear phrases with circular logic or wording. 

Author Response

Dear Reviewer,

The authors would like to thank You for your interest in this manuscript and for your effort in this review. In this revised version we applied all the suggestions You pointed out. 

You can find all modifications in the attached PDF file;

The authors are grateful to You for your important suggestions and corrections.

Thanks again.

Reviewer 3 Report

Comments and Suggestions for Authors

Dear authors,

Firstly, I would like to congrat you for the work. I would like to suggest some things I miss in this manuscript:

1. There are figures that are not referenced along the text

2. Which high-accuracy methods are used to set up the control points around the building area? Where are those points located? How is designed this control points net? Can be shown a plan with the location of them? What kinf of marker is used?

3. Figure 2 shows the control network but it has low resolution. Could it be changed for another one in which the control network is shown in a better way?

4. The acronym CP is used to refer to Control Points. It must be specified the first time thi acronym appears in the text.

5. Avoid the use of we along the text, it must be impersonal.

6. A general view of the building is missed. It would be advaisable to introduce the building, explaining the historical and architectural context, and the configuration of the building.

7. There are two parameters "Fi" and "fi". I would be necessary to explain what they mean.

8. When the registers that are the worst aligned are explained, it would be advaisable to show which are C05 and A02 and give reasons why this bd alignement is produced. Furthermore, a graph in which connections between scans are schematized would be useful.

9. I dont understand what Figure 8 represents

10. To see clearly how the ICP influences, it should be made a graphics in which the curves are superimposed

11. The sentence in line 232 in incomplete

12. In the introduction, it is missed more references to similar works

13. Avoid the use of questions along a scientific text

14. When the TLS data is combined with the photogrammetric data, how is the color corrected?
15. Figure 24 shows Rhino software. Can be said how long it takes to model the wooden beams? 

16. The last part about HBIM it is not well explained.

Thank you very much. Best regards.

Author Response

Dear Reviewer,

The authors would like to thank You for your interest in this manuscript and for your effort in this review. In this revised version we applied all the suggestions You pointed out:

  1. Figures were now referenced along the text
  2. More details are now provided as concern your comments. In particular, the phrase that describes the Control Point Network has is now “The Control Points (CPs), positioned in redundant numbers around the building area and used as reference points for all subsequent survey work, are stainless steel blocks stamped with the point's name.”
  3. A more readable one has replaced Figure 2
  4. CP acronym fixed
  5. the use of We was deleted
  6. the 1.1 chapter was now included concerning the general historic overview of the castle
  7. The statistics fi and Fi (relative frequency and cumulative frequency) have been defined explicitly
  8. The possible reasons for the detected misalignment in C05-A02 are described in the three points bulleted list. A plan of scan station and their connections has been added.
  9. Caption improved in figure 8
  10. ICP influence is in Figure 4 (now Figure 5), where fi and Fi before and after ICP refinements are superimposed
  11. sentence fixed;
  12. references to similar works are now included
  13. questions were removed;
  14. No colour correction was performed.
  15. The modelling process, carried out via Trimble RealWorks, was better explained as concerned time duration
  16. The hbim paragraph was implemented and better explained

The authors are grateful to You for your important suggestions and corrections.

Thanks again.

Round 2

Reviewer 2 Report

Comments and Suggestions for Authors

Good responses to suggested edits and clarifications. 

Author Response

The authors are grateful to You for your important suggestions and corrections.

Thanks again for your contribution. 

Reviewer 3 Report

Comments and Suggestions for Authors

Dear authors, 

In my opinion, although information has been added, the work is not well structured. The reading is not clear. I think you must reorganize ideas and present the methodology in a more organized way. Some references that are included are not justified. The epygraph "The context" i think must be changed fot another title, as this sound really general. Please, take care of what you are writing, and how you are presenting your work. Don't forget you are writing for a scientific public, so the text must be clear, well organized, the mehotds must be explained in a simple but complete way, the different instruments that are used must be showed in a specific epygraph, and the organization of the work must be clear. After having read it twice, I can see a great effort, and a lot of work, but it is not presented in the proper way for a scientific journal. Take your time to re-write it and organize ideas. 

Best regards,

Comments on the Quality of English Language

In general, you must rewrite this work and organized properly the ideas.

Author Response

Dear Reviewer,

The authors would like to thank You again for your second-round review.  Your interest in this manuscript was appreciated by the authors. Also in accordance with the first-round revisions, in this new revised version we applied all the suggestions You pointed out:

  • References have been checked, now resulting relevant to the text and arguments;
  • The title of 1.1 chapter was replaced from “the context” to a more specific “the case study: the medieval castle of Frinco”;
  • A dedicated chapter to surveying equipment has been included (2.1.);
  • The overall structure of the manuscript has been improved;
  • English was checked and revised.

The authors are grateful to You for your important suggestions and corrections for both revision rounds.

Thanks again.